# OpenReview forum: "RAG Makes Guardrails Unsafe? Investigating Robustness of Guardrails under RAG-style Contexts"
_ICLR.cc/2026/Conference — Submitted to ICLR 2026_

### Official Review · Reviewer_r7jG · 2025-10-28

**Soundness:** 3
**Presentation:** 3
**Contribution:** 3
**Rating:** 6
**Confidence:** 2

**Summary:**

This paper proposes an investigation into the robustness of LLM-based guardrails under RAG-style contexts, aiming at addressing the vulnerability of external LLM-based guardrails to data distribution shifts. The study introduces a case study approach using RAG, a systematic evaluation of 3 Llama Guard models and 2 GPT-oss models, and a separate analysis of the effects of each component in the RAG-augmented context. Experimental results confirm that inserting benign documents into the guardrail context alters the judgments of input and output guardrails in around 11% and 8% of cases respectively.

**Strengths:**

1. This work focuses on LLM-based guardrail robustness under RAG contexts. It is well-motivated.
2. The proposed Flip Rate—quantifying guardrail judgment flips between vanilla and RAG-augmented settings without ground-truth—is inspiring and provides a scalable tool for evaluating context robustness.
3. The paper’s figures are clear, effectively visualizing key findings and enhancing interpretability.

**Weaknesses:**

1. In Section 5.2, the rationale for claiming content safety is consistent between vanilla and RAG-augmented settings is unclear. For example, inputs/outputs are safe in the vanilla setting; if RAG-augmented version query violates guardrails’ safety principles, this flip reflects guardrails’ correct judgment rather than poor robustness.

**Questions:**

1. Can you verify on the dataset that such flips stem from guardrails’ correct judgment (not poor robustness)? Reporting its proportion would further justify the experimental setup’s rationality.
2. What is a more robust way to use guardrails models between using them with RAG context or without?

---

> ### Author Response · Authors · 2025-11-25
> **Reply to Reviewer r7jG**
>
> We thank the reviewer for recognizing the novelty of our work and the utility of the Flip Rate metric for scalable evaluation.
> ## On problem definition of guardrail robustness under RAG context (weakness #1)
> Sec. 5.2 studies how RAG-style context affects guardrails’ judgments on safe user queries and their LLM responses, and does not state that content safety is consistent between vanilla and RAG-augmented settings. We suspect the reviewer might be referring to the assumption in Sec. 3.2.
>
> The review asks whether the RAG-augmented version of safe query violates guardrails’ safety principles. We agree that if we insert adversarial documents which involve misinformation or malicious instructions, a normal safe user query would likely become unsafe; a perfect guardrail would be expected to flip its judgment accordingly.
>
> However, the paper is intended to evaluate whether LLM-based guardrails flip the safety judgments when **retrieved context is appended without altering the underlying safety label** of the query/response. Therefore, we chose to retrieve documents from English Wikipedia, which acts as a source of generally benign, factual information. These documents do not contain harmful content, misinformation or malicious instructions to fundamentally alter the safety status of a query/response.
>
> ## Question about Flip Rate (question #1)
> Please see the common reply.
>
> ## Question “What is a more robust way to use guardrails models between using them with RAG context or without?” (question #2)
> The paper investigates whether guardrails are robust against RAG-style context. The robustness is measured by comparing the safety judgments of guardrails between the vanilla setting and RAG-augmented setting. Our results indicate that RAG context is likely to negatively impact the accuracy of the guardrail under unsafe queries, and thus, it is recommended that guardrails be used without RAG until effective mitigations are developed in further research.

---

### Official Review · Reviewer_hda1 · 2025-10-29

**Soundness:** 3
**Presentation:** 3
**Contribution:** 2
**Rating:** 4
**Confidence:** 4

**Summary:**

This paper study how well external LLM-based guardrail models stay consistent when given extra context from RAG. The main question is whether adding retrieved documents changes the guardrail’s safety decisions compared to judging the query or response alone.
The authors propose a metric called Flip Rate to measure how often these safety judgments change. They test both input guardrails and output guardrails using harmful and safe queries. Results show that context causes decision flips in about 11% for input guardrails and 8% for output guardrails. The paper also studies how document relevance and count affect the results and tests simple fixes like prompting or using high-reasoning modes.

**Strengths:**

- The paper studies an important and less explored issue in LLM safety — how stable external guardrails.

- Evaluation are detailed and comprehensive, considering many aspects.

- It presents a simple and useful metric called FR to measure how often guardrail decisions change when the context shifts.

- Experiments show that even normal RAG context can strongly influence guardrail decisions. This reveals a real and practical weakness in current systems.

**Weaknesses:**

- The paper only test BM25 retrieval on Wikipedia. The conclusion may change for dense retrievers or new RAG methods.

- FR only indicates changes in judgment. Change can be good or bad. But this paper regard flips as safety failures sometimes, but some flips may fix mistakes and is useful.

- The prompting and high-reasoning mode fixes are simple and give little improvement.

- The paper only uses normal retrieved documents. It does not test when the documents themselves include adversarial cues or misinformation (knowledge poisoning).

- Missing references

[1] TrustRAG: Enhancing Robustness and Trustworthiness in Retrieval-Augmented Generation

[2] FilterRAG: Zero-Shot Informed Retrieval-Augmented Generation to Mitigate Hallucinations in VQA

[3] Thinking in a Crowd: How Auxiliary Information Shapes LLM Reasoning

[4] A Survey on LLM-as-a-Judge

**Questions:**

1. See weakness

2. Could you explain more about Flip Rate? While it measures inconsistency, how can we be sure a "flip" represents a degradation in safety rather than, in some cases, a context-aided correction?

3. This paper focuses on benign documents. How do these guardrails would perform if the retrieved documents themselves contained adversarial content?

---

> ### Author Response · Authors · 2025-11-25
> **Reply to Reviewer hda1**
>
> We thank the reviewer for recognizing the practical importance of our work and the utility of the proposed Flip Rate metric.
>
> ## Selection of BM25 retriever (weakness #1)
> Please see the common reply.
>
> ## Question about Flip Rate (weakness #2 and question #1)
> Please see the common reply.
>
> ## On the selection of mitigation methods (weakness #3 and question #2)
> We selected "high-reasoning mode" and "enhanced prompting" to establish a baseline for how standard LLM capabilities handle this issue. The goal of RQ3 was to investigate whether general-purpose enhancements could resolve guardrail instability directly. The fact that these widely used methods yield only marginal improvements is a significant finding in itself. It highlights that the instability of LLM-based guardrails to RAG-style context is a non-trivial problem. This underscores the need for in-depth investigation on specifically tailored mitigation techniques suggested in our conclusion.
>
> ## Impact of retrieved document safety (weakness #4)
> The paper is intended to evaluate whether LLM-based guardrails flip the safety judgments when **retrieved context is appended without altering the underlying safety label** of the query/response. Appending adversarial documents that involve misinformation or malicious content can change the safety label of the context. For example, inserting a document containing a jailbreaking instruction of illegal activity would turn a normal safe user query into an unsafe query.
>
> Therefore, our work targets the cases of retrieving benign documents from Wikipedia. The insertion of benign documents only enhances the context with correct information, but does not affect the harmfulness of user query or LLM responses. For example, as stated in the prior work [1], a harmful question is still unsafe in the RAG setting and should never be answered. Surprisingly, even when the added documents are benign, we still observe nontrivial negative flips rates — contrary to the intuitive expectation that adding only safe content should not negatively impact the guardrails.
>
> ## Adding reference (weakness #5)
> Thanks for suggesting these relevant references. The "A Survey on LLM-as-a-Judge"[2] suggests that higher reliability is critical for LLM-as-judges of content safety, echoing the practical importance of our robustness evaluation on LLM-based guardrails. We will integrate these references in the revised version.
>
> [1] RAG LLMs are Not Safer: A Safety Analysis of Retrieval-Augmented Generation for Large Language Models (An et al., NAACL 2025)
>
> [2] A Survey on LLM-as-a-Judge (Gu et al., arXiv 2024)

---

> ### Comment · Reviewer_hda1 · 2025-11-25
>
> Thanks for your reply. Please revise and update your paper based on the responses, and then I will conduct a further review.

---

### Official Review · Reviewer_QBPT · 2025-11-01

**Soundness:** 2
**Presentation:** 3
**Contribution:** 2
**Rating:** 4
**Confidence:** 5

**Summary:**

This paper investigates the robustness of LLM-based guardrails under retrieval-augmented generation contexts. The authors introduce the flip rate to measure how often a guardrail’s safety judgment changes when benign retrieved documents are added. Using five guardrail models across both harmful and safe queries, the study finds that judgments flip in both input-guardrail and output-guardrail cases. Additional analyses isolate the impact of document number, relevance, query safety, and generation model, and two mitigation attempts, high-reasoning mode and RAG-aware prompting, provide only marginal improvements. The work exposes a context-robustness gap in current guardrail systems and calls for training and evaluation frameworks that are robust to retrieval composition.

**Strengths:**

-	The paper identifies and formalizes an interesting failure mode of guardrail models, bridging research on RAG safety and LLM moderation.
-	The evaluation covers diverse guardrails, realistic datasets, and controlled RAG setups. The decomposition across context factors is systematic and insightful.
-	The paper is well structured with clear research questions, consistent definitions, and informative figures.

**Weaknesses:**

-	This paper presents interesting findings but provides little in-depth analysis on why these flipped prediction happens. Existing work [1] also investigates the robustness/reliability of guardrail models from the aspect of prediction uncertainty, which I believe is related to the flipped prediction. Those flipped predictions may display high uncertainty and therefore could be easily manipulated with retrieved documents. I think this could be easily verified and at least discussed in the paper.
-	In addition, the RAG-style queries/responses with the larger context might also explain the weak robustness of the safety prediction, which could act like out-of-domain samples and may not appear in the training data of existing guardrail models. Even though the training data of Llama-Guard is not open-source, existing advanced guardrail models like WildGuard [2] have open-source training data, making it easy to verify them by just comparing the sequence length between the training data and RAG-style queries.
-	Another confounder could be the safety judgment of guardrail models on the documents themselves. These documents/corpora are assumed to be safe with the prior knowledge. However, guardrail models may classify some documents as unsafe due to certain unsafe words/phrases in the doc, or failed prediction of guardrail models. In this case, the guardrail models may make opposite predictions once these documents are sampled. I would recommend an ablation study for this factor.

[1] On calibration of LLM-based guard models for reliable content moderation. ICLR 2025.

[2] Wildguard: Open one-stop moderation tools for safety risks, jailbreaks, and refusals of llms. NeurIPS  2024.

**Questions:**

-	Note that the reasoning model only displays a less than 5% flipped rate. Then, how about other metrics like F1 score and FNR? Does the reasoning model improve the classification performance? Is it the case that the reasoning model corrects the wrong prediction so the flipped prediction happened?

---

> ### Author Response · Authors · 2025-11-25
> **Reply to Reviewer QBPT**
>
> We thank the reviewer for recognizing the novelty of our problem formulation and the systematic design of our evaluation.
>
> ## Impact of guardrail uncertainty and guardrail’s safety judgment of retrieved document (weakness #1,3)
> Thanks for suggesting these two experiments to investigate why flips of guardrail judgment happen. We agree that both the guardrail prediction uncertainty and the guardrail’s safety judgment of the additional documents might be potential factors. Our work is the first step studying the robustness of LLM-based guardrails against RAG-style context. These points are worthy to explore as future work and beyond the scope of this paper.
> ## Impact of sequence length (weakness #2)
> Sequence length is a potential factor degrading LLM performance. Although we didn’t compare the statistics of sequence length of guardrail training data with RAG-style context, we studied the effect of the number of retrieved documents in Sec.5.1.1, which could be a proxy of sequence length.
>
> We varied the number of retrieved documents (k=1, 3, 5, 8, 10). If length were the main driver, we would expect FR to increase significantly as k increases. Instead, we observed that FRs drops significantly at k=1 but then fluctuates slightly as k increases to 10. This indicates that the presence of retrieved documents impacts the guardrail more than the raw quantity of tokens.
>
> ## Question about the effect of reasoning models (question #1)
> As we don’t have safety labels for outputs, here we only discuss the FNRs and F1 scores of input guardrails. Our results show that the RAG-style context **increases FNRs and decreases F1 scores of both reasoning models**.
>
> |  | Non-RAG | k=1 | k=3 | k=5 | k=8 | k=10 |
> | - | - | - | - | - | - | - |
> | GPT-oss-20b     | 9.39% | 10.72% | 10.91% | 10.79% | 11.26% | 11.62%
> | GPT-oss-120b     | 7.56% | 9.05% | 8.50% | 8.71% | 9.13% | 9.10%
> | GPT-oss-120b high-reasoning | 6.55% | 6.93% | 6.96% | 6.72% | 6.99% | 6.88%
>
> The table above displays the FNRs of reasoning models serving as input guardrails.
>
> * **RAG-style context increases the FNR** of both reasoning models gpt-oss-20b (avg. increase of 1.67%) and 120b (avg. increase of 1.34%) when checking safety of input queries.
> * When gpt-oss-120b is configured to **high reasoning effort**, it has the lowest FNR across all k, indicating **reasoning can improve guardrail performance**. However, even in this mode, the FNR still increases at the presence of RAG-style context (avg. increase of 0.3%), confirming that reasoning alone does not fully mitigate the instability.
>
> |  | Non-RAG | k=1 | k=3 | k=5 | k=8 | k=10 |
> | - | - | - | - | - | - | - |
> | GPT-oss-20b     | 80.95% | 77.85% | 74.91% | 75.53% | 74.63% | 75.06%
> | GPT-oss-120b     | 82.58% | 80.96% | 81.19% | 80.70% | 80.26% | 79.79%
> | GPT-oss-120b high-reasoning | 82.89% | 81.57% | 81.90% | 81.05% | 81.63% | 80.81%
>
> The table above displays the F1 scores of reasoning models serving as input guardrails.
> * **RAG-style context reduces the F1 scores** of both reasoning models gpt-oss-20b (avg. drop of 5.40%) and 120b (avg. drop of 2.00%) when checking safety of input queries.
> * Similarly, ppt-oss-120b configured with **high reasoning effort** have highest F1 across all k, but F1 still drops at the presence of RAG-style context (avg. drop of 1.49%)
>
> As we discussed in the common reply, the new flip categorization results show that (1) when checking unsafe queries, most flips of the reasoning models are negative (correct → incorrect); (2) when checking safe queries, most flips of the reasoning models are positive (incorrect → correct). Overall, both reasoning models regress on a significant amount of user queries/LLM responses that they could previously handle correctly.

---

### Official Review · Reviewer_AgfF · 2025-11-04

**Soundness:** 3
**Presentation:** 3
**Contribution:** 3
**Rating:** 4
**Confidence:** 5

**Summary:**

This paper investigates the robustness of safety guardrails when integrated with Retrieval-Augmented Generation (RAG) systems. The authors argue that retrieved documents—while improving factuality—can inadvertently alter guardrail decisions, compromising safety consistency. To quantify this, they propose a label-free metric called Flip Rate (FR), measuring how often a guardrail’s safety judgment changes under varying retrieval conditions. The study evaluates five popular guardrail models (three Llama Guard variants and two GPT-based ones) across both input-level and output-level settings, using thousands of harmful and safe queries retrieved via BM25 from Wikipedia. Extensive experiments reveal that guardrails exhibit notable instability—about 10.9% FR for inputs and 8.4% for outputs—and that factors such as the number and relevance of retrieved documents, query safety level, and the generator model significantly influence robustness. The authors also test lightweight mitigation strategies, including “high reasoning” and “RAG-aware” prompting, which yield only marginal improvements.

**Strengths:**

The paper formalizes guardrail robustness under RAG by defining consistency requirements for input/output guardrails and introducing a label-free Flip Rate (FR) that quantifies judgment changes; it’s simple, scalable, and clearly distinguished from accuracy. It covers 5 guardrails (3 Llama Guard versions and 2 GPT-oss variants), evaluates both input and output settings, and spans 6,795 harmful queries plus additional safe queries—providing unusually comprehensive coverage for this topic. Key findings—e.g., input FR ≈10.9% and output FR ≈8.4%—are concrete and actionable; the work also shows task-dependent robustness differences across guardrails.

**Weaknesses:**

While the proposed Flip Rate (FR) is an elegant, label-free measure of guardrail robustness, it inherently cannot distinguish between correct and incorrect judgment changes. A flip may indicate either an improvement or a degradation in safety performance, but the metric treats both equally. This limitation weakens the interpretability of FR as a genuine proxy for “safety robustness,” and suggests that additional labeled or human-audited analyses would be valuable for validation.

Moreover, all RAG retrieval experiments rely solely on a BM25 retriever over English Wikipedia. This setup overlooks stronger and more representative retrievers (e.g., dense, hybrid, or cross-encoder–based retrieval). Since the retrieved context strongly drives guardrail flipping behavior, it remains unclear whether the reported instability generalizes to modern retrieval architectures. Including at least one dense retrieval baseline (such as Contriever or DPR) would substantially strengthen the empirical conclusions.

**Questions:**

See weaknesses above.

---

> ### Author Response · Authors · 2025-11-25
> **Reply to Reviewer AgfF**
>
> We thank the reviewer for recognizing the novelty of the proposed label-free metric flip rate and the comprehensive coverage of our evaluation on both unsafe and safe queries.
>
> ## On limitation of Flip Rate (weakness #1)
> Please see the common reply.
>
> ## On selection of retriever (weakness #2)
> Please see the common reply.

---

### Author Response · Authors · 2025-11-25
**Reply to the common concerns**

# Reply to the common concerns
We would like to sincerely thank all reviewers for their valuable and constructive feedback. We are greatly encouraged by the positive feedback from the reviewers, which we summarize below.
* **Novelty and Practical Value:** Reviewers agreed that the paper targets an important, novel issue of existing LLM-based guardrails (Reviewer QBPT, hda1, r7jG)
* **Comprehensive Evaluation:** The reviewers strongly supported our methodology, describing the evaluation as comprehensive, systematic, and diverse (Reviewer AgfF, QBPT, hda1).
* **Proposed Metric:** All reviewers (Reviewer AgfF, QBPT, hda1, r7jG) highlighted our formalization of guardrail robustness under RAG—especially the simple, scalable, label-free Flip Rate (FR) metric—as an inspiring and useful tool that is clearly distinguished from accuracy.
* **Significant Experimental Results:** The concrete empirical results in the paper and systematic decomposition analysis reveal a significant and practical vulnerability in the current guardrail systems (Reviewer AgfF, hda1)
* **Clarity:** The paper was commended for its clear structure, definitions, and informative figures (Reviewer QBPT, r7jG).

In this comment, we provide new analysis and experimental results to address **two common concerns**. The rest of concerns are replied as individual comments to reviewers under each of the reviews.

## Limitation of Flip Rate (common weakness #1)
* In order to provide more insights about the flips, we defined positive (incorrect judgment → correct judgment) and negative flips (correct judgment → incorrect judgment).
* With the known safety labels of input queries, we found that 72% of input guardrails’ flips on checking unsafe queries are negative, while 33% of flips on checking safe queries are negative.
* Through manually labeling 200 sampled flips of output guardrails, we found that most output guardrail flips are negative.
* **Conclusion:** Identifying that a significant proportion of flips are negative, we demonstrate that RAG-style contexts can potentially compromise the reliability of input and output guardrails, causing them to regress on user queries/LLM responses that they could previously handle correctly.
## Selection of retriever (common weakness #2)
* We added new experiments with a dense retriever, Contriever[1], to verify whether the RAG-style context obtained through dense retriever can flip guardrail’s judgment similarly like BM25.
* The experimental results show that FRs caused by Contriever’s RAG-style context have a similar pattern as BM25. Taking the conclusion that higher number of documents increase FRs of guardrails slightly (Sec.5.1.1) as example, the input guardrail GPT-oss-20b’s FR changes from 3.9% to 4.8% from k=1 to k=10 with Contriever while it changes from 4.9% to 6.1% with BM25.
* **Conclusion:** The findings in the original paper can be generalized to dense retrievers.

More details about the new analysis and experiments are provided in separate comments.

[1] Unsupervised Dense Information Retrieval with Contrastive Learning (Izacard et al., 2021 arXiv)

---

> ### Author Response · Authors · 2025-11-25
> **Limitation of Flip Rate (common weakness #1)**
>
> # Limitation of Flip Rate (common weakness #1)
> We thank the reviewers for the insightful critique regarding the Flip Rate (FR) metric. We agree that the proposed label-free robustness metric FR cannot distinguish between "correction" and "degradation".
> To address this, we categorized all flips into two fine-grained definitions:
> * Positive Flip: Incorrect judgment flips to Correct.
> * Negative Flip: Correct judgment flips to Incorrect.
>
> Fewer negative flips and more positive flips are desirable. A smaller number of negative flips indicates that the guardrail can robustly maintain its correct judgment against the context change, while a higher number of positive flips reflects how much incorrectness can be mitigated by the richer context.
>
> We found that a significant proportion of flips are negative for both input and output guardrail. This indicates that RAG-style contexts can compromise the reliability of LLM-based guardrails, causing them to regress on user queries/LLM responses that they could previously handle correctly.
>
> The full data and analysis results of new experiments will be integrated in the revised paper.
>
> ## Input guardrails
> Since the safety labels of input queries are given by the datasets, we categorized all flips of input guardrails. Here, we discuss several important results in the context of each RQ.
>
> ### RQ1 Most flips on unsafe queries are negative
> For unsafe queries (document number k=5), the majority of flips (61%-89%) are negative, averaging 72%. Since there are a greater number of negative flips than positive flips, it can be concluded that RAG-style context degrades the performance of input guardrails.
>
> ### RQ2.1.1 More document could lead to higher negative flip ratio
> Negative flips consistently exceed positive ones across all k and guardrails (except LlamaGuard3 at k=3, ~49% negative). We observed a clear increasing trend in the negative flip ratio for LlamaGuard3 (50.4%→75.1%) and LlamaGuard4 (55.1%→94.7%) when k grows from 1 to 10, showing that additional benign documents in the unsafe query can cause more harm to certain guardrails.
>
> ### RQ2.2 Most flips on safe queries are positive
> In contrast to unsafe queries, the negative flip ratio for safe queries is much lower (10%-57%, avg 33%), mitigating the overall FPRs. Llamaguard 3&4 have a clear decreasing trend in the negative flip ratio when k increases.
>
> Despite the FPR gains, it is undesirable for guardrails to introduce new errors on instances they previously classified correctly (w/o documents). The negative flip ratio of 33% shows that a significant part of original functionality is broken by RAG-style context.
>
> ### Contradictory results between RQ1 and RQ2.2 – Benign RAG biased Guardrails toward "Safe"
> With RAG-style context, input guardrails make more incorrect judgments of "safe" when judging unsafe queries, and make more correct judgments of “safe” when judging safe queries. This suggests that appending retrieved documents leads input guardrails to return more "safe". We have two speculations on why this happens.
> * First, the benignity of the retrieved documents from wikipedia might bias the model towards "safe".
> * Second, the RAG-style context might introduce noises leading to more uncertainty that defaults the model to a “safe” output.
>
> While this reduces False Positives on safe queries, it severely compromises safety by masking unsafe content (higher FNR).
>
> ### RQ3 High-reasoning reduces negative flips but dedicated prompting increases
> High-reasoning mode can effectively lower negative flip ratio of gpt-oss-120b from 67.7% to 53.0% (k=5) in addition to reducing FR (original finding). However, negative flips remain the majority.
>
> Surprisingly, while dedicated prompting reduces FRs across 3 guardrails (original finding), it increases the negative flip ratio for 4 guardrails (avg. increase of 5.8% for these 4 guardrails, k=5).
>
> This reinforces our conclusion that general enhancements are insufficient to fix RAG-induced vulnerability. More research on mitigating the guardrail negative flips is needed.
>
> ## Output guardrails — Most flips are negative
> As output guardrails lack ground-truth labels for generated responses, we randomly sampled and inspected 30 malicious and 10 benign queries’ LLM responses where the output guardrail flipped its judgment, for each guardrail model. In total we have 200 flips and 46.5% of them are from safe to unsafe.
>
> Among the 200 samples, we found that 63.5% were negative flips. On our limited labeled data set, the RAG-style context lowered output guardrail accuracies. Future work could leverage new labeled benchmarks to investigate the factors impacting the negative flip ratio of output guardrails.
>
> ## Conclusion
> Finding that both input and output guardrails have a substantial ratio of negative flips, we demonstrate that RAG-style contexts can potentially undermine the reliability of guardrails, yielding regressions on queries/responses they had previously classified correctly.

---

> ### Author Response · Authors · 2025-11-25
> **Selection of retriever (common weakness #2)**
>
> # Selection of retriever (common weakness #2)
> We thank the reviewer for highlighting the importance of validating our findings across different retrieval architectures. We agree that relying solely on BM25 could potentially limit the generalizability of our conclusions.
>
> To address this, we have conducted a new set of experiments using **Contriever** to validate the generalizability of our findings. We compared these results against our original results of BM25. The full results will be added to the revised paper, but we summarize the key findings here.
>
> ## Retriever Setup
> We replicated our RAG pipeline using Contriever to generate embeddings (dim=768) for the same Wikipedia corpus used in the main paper. We employed FAISS (IndexFlatIP) for similarity search to retrieve the top-k documents.
>
> ## Input Guardrails
> Contriever lowers all guardrails’ FRs across all k for both unsafe and safe queries, but there still remains a significant number of flips and the trend of FRs with Contriever closely mirrors that of BM25.
>
> For examples,
> * **Conclusion of Sec.5.1.1:** Introducing even a single retrieved document in the context significantly alters guardrail judgments, but additional documents contribute little incremental harm. The table below displays the FRs of GPT-oss-20b checking unsafe queries with the two retrievers across different k. We can see that when k=1 both retrievers lead to non-trivial number of flips, and their FRs are increased a little bit by additional documents.
>
> |  | k=1 | k=3 | k=5 | k=8 | k=10 |
> | - | - | - | - | - | - |
> | BM25     | 4.91%     | 5.24% |5.52% | 5.64% | 6.11% |
> | Contriever     | 3.96%     | 4.46% |3.95% | 4.82% | 4.83% |
>
> * **Conclusion of Sec 5.2:** Guardrails are not robust to RAG-style context perturbation even on safe queries. The table below displays the FRs of 5 guardrails checking safe queries with the two retrievers when k=5. It is obvious that both retrievers have the same relative ranking of guardrails with regard to FRs ( Llama Guard 4 < GPT-oss-120b < Llama Guard 3 < GPT-oss-20b < Llama Guard 2).
>
> |  | Llama Guard 2 | Llama Guard 3 | Llama Guard 4 | GPT-oss-20b | GPT-oss-120b |
> | - | - | - | - | - | - |
> | BM25     | 21.02%     | 13.83% | 7.01% | 17.39% | 11.02% |
> | Contriever     | 17.39%     | 13.25% | 6.18% | 14.94% | 9.36% |
>
> ## Output Guardrails
> We observed that Contriever brings similar changes on FRs for output guardrail.
>
> The table below shows the FRs of 5 guardrails judging responses to unsafe queries with the two retrievers when k=5. Although Contriever reduced the FRs of 4 guardrails, there remains a non-trivial number of flips. Two retrievers also have similar relative rankings of guardrails with regard to FRs (except for GPT-oss-120b).
>
> |  | Llama Guard 2 | Llama Guard 3 | Llama Guard 4 | GPT-oss-20b | GPT-oss-120b |
> | - | - | - | - | - | - |
> | BM25     | 3.96%     | 5.40% | 3.86% | 11.18% | 4.67% |
> | Contriever     | 3.33%     | 4.80% | 3.06% | 7.68% | 4.83% |
>
> ## Conclusion
> These new experiments confirm that the **findings reported in our paper can be generalized to modern dense retrieval architectures**. While a dense retriever (Contriever) mitigates the instability slightly, it does not solve it, reinforcing the need for more investigation in guardrail robustness and improvement techniques.

---

### Author Response · Authors · 2025-11-29
**Revised paper update**

We have just uploaded a revised version of our paper. This version reflects the changes discussed in our rebuttal responses, e.g. the new fine-grained analysis of flip rate, new experiments about the selection of retriever and new citations.

The changes are highlighted in blue for easier tracking. We believe these updates address the concerns raised and further strengthen the paper's contribution. Thank you again for the constructive feedback that helped improve the work.

---

### Meta-Review · Area_Chair_5QB8 · 2026-01-06

**Summary:**

This submission investigates the robustness of LLM-based guardrails under RAG-style contexts and introduces Flip Rate as a metric to quantify judgment instability. Reviewers generally agree that the topic is relevant and timely. However, multiple reviewers consistently raised concerns regarding the methodological soundness and interpretability of the proposed evaluation.

Reviewers consistently raised concerns regarding the interpretability and validity of the proposed Flip Rate metric. The authors have taken steps to address this by distinguishing between positive and negative flips, which helps clarify the empirical observations. However, the revision does not fully resolve the underlying ambiguity as to whether judgment flips under RAG-style contexts should be interpreted as genuine robustness degradation, since the work still lacks a principled ground-truth definition of safety in such settings.

Furthermore, reviewers noted a lack of in-depth investigation into why RAG-style contexts induce judgment flips.  The paper would benefit from a more in-depth analysis that disentangles the potential factors contributing to the observed instability, such as distribution shift, increased context length, model uncertainty, or the guardrails’ treatment of retrieved documents themselves. While the additional mitigation experiments and expanded discussion are informative, they mainly strengthen the paper’s position as a systematic characterization of the phenomenon, rather than providing deeper explanatory insights or methodological advances into the root causes of the issue.

Given that these concerns relate to the paper’s core methodology and remain insufficiently resolved, and considering the overall reviewer scores, I recommend rejection.

**Reviewer Concerns:**

The reviewer's concerns may be addressed by the rebuttal:
- For the FR metric limitation, the authors categorized flips into positive and negative types, confirming that most flips are negative (degrading guardrail performance) through labeled analysis of input and output guardrails.
- For retrieval architecture generalizability, the authors added experiments with the dense retriever Contriever, showing that FR trends are consistent with BM25, confirming the findings’ generalizability.

The reviewer's concerns may still be outstanding:
- While FR is categorized, the metric itself remains unable to inherently distinguish flip types, requiring additional labeled data for interpretation—limiting its scalability as a label-free tool.
The study still does not address adversarial retrieved content (e.g., misinformation, jailbreaking cues), leaving uncertainty about guardrail performance in high-risk RAG scenarios.
- Key flip causes (e.g., guardrail uncertainty about retrieved documents, training data sequence length mismatches) are identified as future work but not empirically verified in the current study.
- Mitigation strategies remain ineffective, with no targeted solutions proposed to address the core context-robustness gap.
- Experimental metrics are still incomplete: output guardrails lack ground-truth labels for comprehensive FNR/F1 analysis, and LLM-as-a-judge evaluations (e.g., GPT-4) are not included to validate semantic safety.

**Reviewer Scores:**

- For Reviewer AgfF, he may not change their score, as core concerns about FR interpretability and dense retriever validation are partially addressed, but the metric’s inherent limitation and lack of adversarial content testing persist.
- For Reviewer QBPT, he may not change their score, as key concerns about flip causes and sequence length impacts are not fully verified, and mitigation effectiveness remains limited.
- For Reviewer hda1, he may not change their score, as outstanding concerns about adversarial content, mitigation inadequacy, and reference integration.
- For Reviewer r7jG, he may not change their score, as the authors clarified the experimental setup’s focus on benign documents, but questions about flip validity and lack of robust guardrail usage guidelines remain partially unresolved.

---

### Decision · Program_Chairs · 2026-01-26

Reject